# Is Belonging to a Religious Organization Enough? Differences in Religious Affiliation Versus Self-Ratings of Spirituality on Behavioral and Psychological Variables in Individuals with Heart Failure

**DOI:** 10.3390/healthcare8020129

**Published:** 2020-05-08

**Authors:** Jesús Saiz, Meredith A. Pung, Kathleen L. Wilson, Christopher Pruitt, Thomas Rutledge, Laura Redwine, Pam R. Taub, Barry H. Greenberg, Paul J. Mills

**Affiliations:** 1Department of Social, Work and Differential Psychology, Complutense University of Madrid, 28223 Madrid, Spain; 2Department of Family Medicine and Public Health, University of California San Diego, La Jolla, CA 92093, USA; mpung@ucsd.edu (M.A.P.); k8wilson@ucsd.edu (K.L.W.); cpruitt@ucsd.edu (C.P.); pmills@ucsd.edu (P.J.M.); 3VA San Diego Healthcare System, La Jolla, CA 92161, USA; thomas.rutledge@va.gov; 4Department of Psychiatry, University of California San Diego, La Jolla, CA 92093, USA; 5College of Nursing, University of South Florida, Tampa, FL 33612, USA; lredwine@usf.edu; 6Department of Medicine, University of California San Diego, La Jolla, CA 92093, USA; ptaub@ucsd.edu (P.R.T.); bgreenberg@ucsd.edu (B.H.G.)

**Keywords:** behavioral health, heart failure, well-being, spirituality, religious affiliation

## Abstract

In the United States, heart failure (HF) affects approximately 6.5 million adults. While studies show that individuals with HF often suffer from adverse symptoms such as depression and anxiety, studies also show that these symptoms can be at least partially offset by the presence of spiritual wellbeing. In a sample of 327 men and women with AHA/ACC classification Stage B HF, we found that more spirituality in patients was associated with better clinically-related symptoms such as depressed mood and anxiety, emotional variables (affect, anger), well-being (optimism, satisfaction with life), and physical health-related outcomes (fatigue, sleep quality). These patients also showed better self-efficacy to maintain cardiac function. Simply belonging to a religious organization independent of spiritualty, however, was not a reliable predictor of health-related benefits. In fact, we observed instances of belonging to a religious organization unaccompanied by parallel spiritual ratings, which appeared counterproductive.

## 1. Introduction

Cardiovascular disease (CVD) is the leading global cause of death, accounting for 17.3 million deaths per year, a number that is expected to increase to more than 23.6 million by 2030. In the United States, heart failure (HF) affects approximately 6.5 million adults with an estimated $30 billion annual economic burden and a significant loss of quality of life for these individuals [1].

There is empirical evidence demonstrating the existence of positive associations between spirituality and quality of life [2], and spirituality and better health [3,4,5]. Such associations are seen for both genders for mental health [6] as well as chronic and physical disorders [7]. 

A recent literature review shows these same benefits for individuals with HF [8]. Individuals with HF who report more daily spiritual experiences have higher existential well-being [9]. Mills et al. [10] showed that gratitude statistically mediated spiritual well-being and better sleep quality and less depressive mood. Broadly, this literature indicates that spiritual wellness may be a protective factor against many adverse symptoms seen in HF, having an overall positive impact on the wellness of HF patients [8,11,12].

For the purposes of scientific research, spirituality has been defined by the Joint Commission [13] (p. 7) “as a complex and multidimensional part of the human experience-our inner belief system. It helps individuals search for the meaning and purpose of life, and it helps them experience hope, love, inner peace, comfort, and support, being the experiences of meaning in life and connectedness, spirituality’s central elements.” In contrast, religiosity is understood as participation in the institutionally sanctioned beliefs and activities of a particular faith group [14]. Religious attendance and religious affiliation have been both associated with better health outcomes [15]. Yet, not everyone that defines themselves as being spiritual belongs to a religious institution, nor is belonging to a religious institution a requirement for spirituality [16]. While studies emphasize the benefits of spirituality for better health [17,18], including HF [8,10,11], it remains unknown whether belonging to a religious institution without a spiritual component, is sufficient to obtain the same benefits in HF patients. In fact, in their longitudinal study in a national sample of African Americans, Roth and colleagues [19] concluded that “simply attending religious services will not, by itself, achieve health benefits” (p. 423), pointing to the necessity to consider the role of other variables. 

To address this issue, the present study examined relationships of religious affiliation and spirituality on a set of psychological and physical symptom measures in asymptomatic HF patients. The study addressed the following two aims:(1)Identify potential relationships of religious affiliation and spirituality on psychological and physical health symptoms in HF patients.(2)Assess the presence of interactions between religious affiliation and spirituality, and determine if belonging to a religious institution alone is enough to improve psychological and physical health symptoms.

## 2. Materials and Methods

### 2.1. Sample 

As it is described in Table 1 and Table 2, the sample consisted of 327 people with AHA/ACC classification Stage B HF (312 men and 15 women), with a diagnosis for at least 3 months. Participants were recruited from the cardiology clinics at University of California San Diego (UCSD) and at the Veterans Affairs San Diego Healthcare System (VASDHS) between 2011–2014. The presence of Stage B HF was defined as structural heart disease based on recommendations and cut-points from the American Society of Echocardiography guidelines [20,21].

### 2.2. Study Design and Procedure

We conducted a cross-sectional study to examine the primary and interaction effects of spirituality and religious affiliation on an extensive panel of psychological and physical health variables. These variables were specifically selected because, as was reviewed above and it is detailed in the next section, they have already shown their importance for HF in previous studies, and also they have proved to be very sensitive to describe different outcomes in religion/spirituality and health researches. The protocol was approved by the UCSD and VASDHS Institutional Review Boards (#101194). Participants gave written informed consent. The study was carried out in accordance with the Declaration of Helsinki principles.

### 2.3. Measures

#### 2.3.1. Religious Affiliation

To assess this issue, we selected item number 7 of the Social Network Index [22], which specifically asks: “Do you belong to a church, temple, or other religious group?” “Yes” was decoded as 2 and “No” as 1. Although religiosity is a complex phenomenon, and it comprehends other indicators as religious attendance or religious practices, religious affiliation has been typically used by other authors on religion and health researches [15]. 

#### 2.3.2. Spirituality 

To assess spirituality, the meaning/peace subscale of the FACIT-SP12 Questionnaire was chosen, because it integrates elements (meaning, purpose, harmony, peace, connection) that have been traditionally considered in the definition of spirituality (e.g., “I feel a sense of harmony within myself”) [23]. In addition, Peterman and others [14], found that this subscale was not significantly associated with existing religion measures, suggesting that what it reports is independent of religion. The range of responses is 0–32. As it is explained latter, for the design purposes, responses were dichotomized, generating two groups: low in spirituality (scores between 0 and 16), and high in spirituality (scores between 17 and 32). We chose 16 as the cut point to make it coincide with the middle of the response rank. 

#### 2.3.3. Clinical Dimensions

Symptoms of depression were assessed with the 21-item Beck Depression Inventory (BDI–1A), where scores ≥10 indicate possible clinical depression [24]. The BDI–1A assesses symptoms in two dimensions and one global score. The cognitive-affective dimensions include negative mood or negative affect, while the somatic dimension includes symptoms such as fatigue or loss of energy. The BDI–1A shows high reliability and structural validity and capacity to discriminate between depressed and nondepressed participants with broad applicability for research and clinical practice worldwide [25], including the research area of spirituality and health [26]. 

Anxiety symptoms were assessed using the State-Trait Anxiety Inventory (STAI) [27]. It is divided into two parts: state anxiety and trait anxiety, with the former considered more temporary anxiety symptoms and the latter a more general and long-standing disposition towards anxiety. Each part consists of 20 descriptive assertions, and answers are given in a four-point Likert scale (1 = not at all to 4 = very much so). Scores of each part range from 20 to 80 points, and indicate either a low degree of anxiety (0–30), a moderate degree of anxiety (31–49), or a high degree of anxiety (equal to or over 50). The STAI has been used to study the relationship between anxiety and spirituality (e.g., [28]).

#### 2.3.4. Affect and Anger

The 20-item version of the Positive and Negative Affect Schedule was used to assess affect dimension (PANAS [29]). The items consist of adjectives describing each affective symptom and are rated for how much they were experienced in general on a 5-point Likert scale (1 = not at all and 5 = extremely). Words such as “interested”, “strong”, and “inspired” measure positive affect, and words such as “guilty”, “afraid”, and “hostile” measure negative affect. The scale has been related to spirituality in several studies (e.g., [30]). 

The state-trait anger expression inventory (STAXI) [31] was used to assess how anger is experienced, expressed, or controlled. The STAXI contains 44 items that are rated on a Likert scale of 1–4. The STAXI subscales include state anger, trait anger, anger expression-in (AX-IN; suppression of anger; “I keep things in”), anger expression-out (AX-OUT; externalization of anger towards others or the environment; “I lose my temper”), anger control (AX-CON; “I control my angry feelings”), and anger expression index (AX/EX). AX/EX scale provides a single-number score that estimates the frequency and likelihood of expressing anger (whether to self or to others), the score ranges from 0 to 72 (lower scores reflect less anger expression). In this paper, we focused on anger externalization and control only, and did not assess state or trait anger. We therefore used 24 items instead 44. Anger expression has shown to be useful in other spiritual and health research (e.g., [32]).

#### 2.3.5. Well-Being

To examine well-being, we choose the life orientation test-revised (LOT-R) [33] and the satisfaction with life scale (SWLS) [34], as they probe well-being from different perspectives on spirituality and health (e.g., [35,36]). The LOT-R consists of 10 items, three items assess optimism (e.g., “I am always optimistic about my future”), three items assess pessimism (e.g., “If something can go wrong for me, it will”) and there are unscored four filler items. On a 5-point Likert scale, response categories ranged from 1 = strongly agree to 5 = strongly disagree. The scores of the optimism and pessimism sub-scales are the sum of the scores of the corresponding items. A total score was calculated, adding the optimism and the inverted pessimism score. The SWLS is a Likert-like questionnaire with 5 items that are evaluated on a 7-point scale ranging from 1 = strongly agree to 7 = strongly disagree). The 5 items assess current life status compared with the ideal, conditions of life, satisfaction with life, gains in life, and desired changes with life (e.g., “In most ways my life is close to my ideal”. The final score was calculated adding the 5 items.

#### 2.3.6. Physical Symptoms Sleep and Fatigue

The multidimensional fatigue symptom inventory–short form (MFSI) was used to assess fatigue [37]. It contains 30 items based on a 5-point Likert scale (where 1 = not at all and 5 = extremely), which describe how responders feel in six domains (general, physical, emotional, mental, total fatigue, and one dimension to evaluate the self-perception of the responder’s vigor) (e.g., “I feel energetic”). The MFSI has strong psychometric properties and is useful in medically ill and nonmedically ill individuals [38]. 

The Pittsburgh sleep quality index (PSQI) was used to assess sleep quality [39]. The PSQI is widely used in sleep research and measures sleep disturbance and sleep habits, and has high internal reliability and construct validity [40]. We used the following PSQI domains: subjective sleep quality, latency and duration, habitual sleep efficiency, sleep disturbances, and daytime dysfunction, as well as the PSQI global score [41]. We reversed coded the PSQI global score (rank 0–21), so the higher values means better quality of sleep. 

#### 2.3.7. Cardiac Self-Efficacy

The cardiac self-efficacy scale (CSES) examines the role of patient self-efficacy for people with heart disease. The CSES has two factors (control symptoms and maintain function) and has high internal consistency and good convergent and discriminant validity [42]. We used the CSES-Maintain Function subscale, which includes 5 items on a Likert scale, scored from 1 = not at all confident to 5 = completely confident. MFSI, PSQI, and CSES have been successfully used together in previous research on spirituality and health [10].

### 2.4. Data Analysis

Descriptive analyses were initially performed with the measures of religious affiliation, spirituality, and psychological and physical health symptoms (IBM SPSS v26, Armonk, NY, USA). Considering the small representation of women in our sample, we checked if there were any differences in the dependent variables considering sex with Student’s *t* test. Analysis of variance (ANOVA) tests were then used to compare the religious groups (people who belong to a religious organization or not, and people high in spirituality vs people low in spirituality). The distribution of spirituality and religious affiliation in the basic sociodemographic and clinical data of the patient sample were also tested to find any potential confounding factors in the analyses. When the Levene test reported non-compliance with the assumption of homogeneity, the robust Brown–Forsythe test was used for group comparisons. 

Secondly, two-factor multivariate analysis of covariance (MANCOVA) and two-factor analysis of covariance (ANCOVA) were performed to determine if there were interactions between spirituality and religious affiliation. Age and body mass index (BMI) functioned as covariate due to the importance of BMI and age for HF patients [43]. When an interaction was significant, simple effect analyses were performed in lieu of main effects. When different sub-scales composed the dependent variable, we used MANCOVA, but when dependent variable was a unique measure, we used ANCOVA. For MANCOVA, the Pillai trace was taken as a criterion for its robustness [44]. Estimates of the effect size were carried out using partial eta-squares, evaluating these effects with the criteria of 0.02, 0.13, and 0.26 as cut-off points for small, medium, and large effects, as has been done in similar studies [45].

## 3. Results

As seen in Table 2, there were no significant differences in the distribution of men and women in the two categories. Considering sex, we only found significant differences on the variables state anxiety (*t*[14.62] = 2.97, *p* < 0.05) and sleep duration (*t*[18.97] = 3.78, *p* < 0.01), so we decided to keep our women in the sample and be cautious with the interpretation of the two variables indicated. On the other hand, we observed differences in the rest of the variables when referring to the two spirituality categories: people high in spirituality showed more positive values than those low in spirituality. There were no differences in relation to the two religious affiliation categories. In addition, we did not find any other significative differences in the distribution of spirituality and religious affiliation in the basic sociodemographic and clinical data of our sample.

### 3.1. Clinical Variables

On BDI-II scores, the multivariate test yielded significant results for the spirituality factor (F[2, 290] = 62,626, *p* < 0.001, *η^2^* = 0.302). Specifically, as observed in Table 3, we found that for the dimensions of cognitive, somatic and total depression, those higher in spirituality scored significantly lower, regardless of whether they belonged to a religious organization.

For the state and trait anxiety scores, the multivariate test yielded significant results for the spirituality factor (F[2, 246] = 69,676, *p* < 0.001, *η^2^* = 0.362), religiosity (F[2, 246] = 5.466, *p* < 0.01, *η^2^* = 0.043), and the interaction between the two (F[2, 246] = 3158, *p* < 0.05, *η^2^* = 0.025). Specifically, there was an interaction between spirituality and religious affiliation on state anxiety. Those people who belonged to an organization but were not spiritual scored higher in state anxiety (Figure 1a and Table 3). For trait anxiety, people who had higher spirituality had lower trait anxiety than those who were not spiritual. Those who belonged to a religious organization scored higher in trait anxiety than those who did not belong. However, the effect size of the variable spirituality was large, while the effect size of the variable religious affiliation was small.

### 3.2. Emotional Variables

On the affect dimension, the multivariate test yielded significant results for spirituality (F[2, 299] = 74.26, *p* < 0.001, *η^2^* = 0.33), religious affiliation (F[2, 29] = 9.699, *p* < 0 001, *η^2^* = 0.06) and their interaction (F[2, 299] = 7.29, *p* < 0.01, *η^2^* = 0.04). As shown in Figure 1b, the interaction for negative affect suggested that people low in spirituality, and who also belonged to a religious organization, had the highest negative affect. For positive affect, however, there was no interaction, and the only significant effect was that of spirituality, indicating that the more spiritual people scored, the higher the positive affect score, regardless of their membership to a religious organization (Table 4).

For anger, the multivariate test yielded significant results for the spirituality factor (F[3, 292] = 26.20, *p* < 0.001, *η^2^* = 0.21) and an interaction between spirituality and religious affiliation (F[3, 292] = 4.78, *p* < 0.01, *η^2^* = 0.04). Specifically, the interaction for anger expression-out, those people who were not spiritual and belonged to a religious organization expressed their emotional experience of anger in an outwardly negative manner (Figure 1c). On the other hand, we found the effect of spirituality in the other three dimensions studied, observing that the more spiritual people scored equally lower in anger expression-in and anger expression index, implying a lower estimate of the person’s tendencies to express anger either outwardly towards other people, or inwardly towards herself. However, they scored higher on anger control, which means they were more able to monitor and control the physical or verbal expressions of anger. These last three results were independent of religious affiliation.

### 3.3. Psychological Well-Being

In the ANCOVAS of LOT-R and SWLS scores, differences were identified for the spiritual factor. We observed that the more spiritual people scored higher in optimism (F[1, 298] = 70.68, *p* < 0.001, *η^2^* = 0.19) and in satisfaction with life (F[1, 116] = 26.97, *p* < 0.001, *η^2^* = 0.18), regardless their religious affiliation.

### 3.4. Physical Health Variables

The multivariate test for the MFSI yielded significant results for spirituality (F[5, 289] = 33.96, *p* < 0.001, *η^2^* = 0.37) and religiosity (F[2, 289] = 5.34, *p* < 0.001, *η^2^* = 0.08). A two-factor ANCOVA indicated a significant interaction effect for the variable emotional fatigue. As can be seen in Figure 1d, people who were low in spirituality and belonged to a religious organization scored highest in emotional fatigue. Physical, mental, and total fatigue were significant for both factors, spirituality and religious affiliation. As observed in Table 4, we found that more spiritual people scored lower in these dimensions than those who were less spiritual, whereas people who belonged to a religion showed more physical, mental and total fatigue than who did not belong to a religious organization. Finally, the more spiritual people also showed more vigor and less general fatigue than the less spiritual.

For sleep, the multivariate test yielded significant results of medium size only for the spirituality factor (F[7, 280] = 8.86, *p* < 0.001, *η^2^* = 0.18): people high in spirituality showed a better quality of sleep, less time to fall asleep, less discomfort during sleep, less trouble staying awake during daily activities, greater efficiency, duration, and better overall test score (Table 5). Alternatively, people who belonged to a religious organization also showed fewer problems staying awake during daily activities than those that did not belong to a religious organization (*p* = 0.003).

Finally, when cardiac self-efficacy was analyzed, the two-factors ANCOVA showed significant effects for the spirituality factor (F[1, 285] = 87.88, *p* < 0.001, *η^2^* = 0.23) and the co-variables age (F[1, 285] = 5.91, *p* < 0.05, *η^2^* = 0.02) and BMI (F[1, 285] = 10.49, *p* < 0.01, *η^2^* = 0.03). Accordingly, people high in spirituality had more confidence in maintaining their cardiac function in daily activities (social, work, family, sexual, and sports) than those with a low profile on spirituality.

## 4. Discussion

This study extends the literature on religion, spirituality and health in HF by addressing the question whether religious organization belonging alone is enough to facilitate better psychosocial and physical health. In a Stage B HF population, we examined the potential differential relationships between spirituality and religious affiliation on an extensive set of behavioral and psychological variables.

The first primary observation, from a multifactorial perspective, is that spirituality alone was strongly and consistently associated with all variables that were studied. Thus, for people with asymptomatic HF, more spirituality was associated with better clinically-related symptoms such as depressed mood and anxiety, emotional variables (affect and anger), well-being (optimism and satisfaction with life), and physical health-related outcomes (fatigue and sleep quality). These patients also showed self-efficacy to maintain cardiac function. Together, these results support some prior studies [10,26,28,30,32,35,36], although they also contradict others [46], for which spirituality was positively related with anger expression.

Spirituality was also shown in the current study to have larger associations with the psychosocial and physical symptom measures than belonging to a religious organization. In our analyses, simply belonging to a religious organization independent of spiritualty was not a reliable predictor of health benefits. In fact, we observed instances where religious affiliation, when it is not accompanied by a parallel spiritual life, appeared counterproductive. For example, in our analyses we observed more negative affect, greater state anxiety, more uncontrolled expression of anger, and greater emotional fatigue among those belonging to a religious organization while endorsing low spirituality. This interaction could be explained by means of the theory of cognitive dissonance [47], which warns that people must maintain coherence between their attitudes, beliefs and behaviors. For instance, if a person belongs to a religious organization but at the same time does not maintain a spiritual life consistent with it, it can generate a cognitive discomfort (dissonance), which in turn might lead to poorer health and well-being. The experiments conducted by Yousaf and Gobet [48] point in this direction, since when the authors examined the emotional consequences of religious dissonance, they found that dissonance induced through religious hypocrisy results in guilt and shame. The necessity of considering the values perspective in clinical settings has been suggested also in other fields like drug addiction [49,50]. Furthermore, Although Allport’s perspective was criticized for its possible conceptual and methodological limitations [51], we might consider also Allport and Ross’ [52] proposal about the difference between intrinsic and extrinsic religion. For these authors, “persons with intrinsic orientation find their master motive in religion”, while “persons with extrinsic orientation are disposed to use religion for their own ends”. Following this reasoning, we might contemplate intrinsic religious orientation people as more spiritual. Moreover, according to our results, different studies have found better outcomes for persons high in intrinsic religiosity than extrinsic religiosity in health related issues as, for instance, suicide [53], depression [54,55] or trait anxiety [56]. 

It is important to recognize that the present study has certain limitations that temper our interpretations of the findings. Because this is a cross-sectional study, we can describe correlations but cannot infer that either spirituality or religious affiliation have causal relationships on the psychosocial or physical health symptoms studied here. The meaning/peace subscale of the FACIT-SP12 questionnaire may have some overlap with other psychosocial instruments that were used. This overlap may explain at least in part the associations we observed between spirituality scores and better psychosocial health. Nevertheless, we believe that this only confirms the need for considering meaning, purpose, harmony, peace, and connection as relevant issues when addressing health problems. In addition, the religious affiliation variable was limited to “church, temple, or other religious group” and was measured dichotomously, which forced people to place themselves in one of the two poles and did not allow possible intermediate ones. For example, we could have missed people who would report “I belong to a religious organization but do not practice the religion”. Religious affiliation might not be the most sensitive option to explore religiosity, and it should be combined with other items as religious practices and religious attendance [15]. Finally, given the specificity of the sample, we cannot generalize to other potential populations, such as those with other physical or mental health conditions.

For future replications of this study, it would be useful to increase the number of participants for each condition, especially for the people low in spirituality that also belonged to a religious organization, which showed great interactions and was small. Perhaps it would be useful to deep in the intrinsic and extrinsic religion concepts, and clarify if extrinsic religion can be spiritual, as well as continue working in a model of spirituality and religion’s effects on health [3], integrating psychosocial and biological variables. In addition, more women should be incorporated in the sample, as two of our dependent variables were affected by sex, and there are important references that highlight gender differences in the prevalence of depression [57], anxiety [58], and other variables regard to epidemiology, diagnostics, or the impact of heart failure on psychosocial factors and healthcare utilization [59].

Individuals with HF have consistently elevated depressed mood and anxiety [21,60], which correlates with significantly poorer clinical outcomes, including morbidity and mortality [61,62]. It may be therefore that for individuals with asymptomatic HF, engaging in spiritual practices (as mindfulness has been recently proposed [63], could help support and improve their clinical symptoms and psychosocial well-being. 

## 5. Conclusions

The present observational study of patients with HF contributes to the growing literature evaluating the value of spirituality for better mental health and quality of life. However, simply belonging to a religious organization independent of spiritualty was not a reliable predictor of health-related benefits. In fact, we observed instances where religious affiliation when it was not accompanied by parallel spiritual ratings, appeared counterproductive. This means that for patients with HF, reinforcing spirituality elements like meaning, purpose, harmony, peace, and connection might help to improve their mental health and quality of life.

## Figures and Tables

**Figure 1 healthcare-08-00129-f001:**
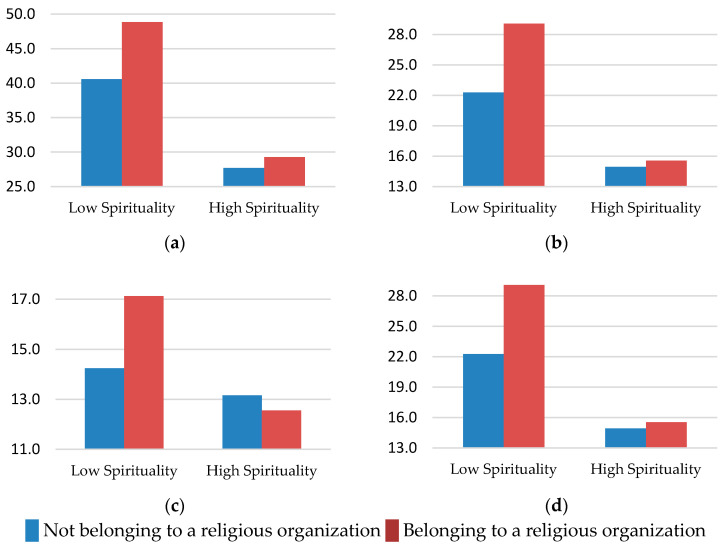
The interaction effect of spirituality and religious organization belonging on the following health variables: (**a**) State anxiety, (**b**) Negative affect, (**c**) Anger expression-out, (**d**) Emotional fatigue.

**Table 1 healthcare-08-00129-t001:** Basic sociodemographic and clinical data of the patient sample.

Marital Status	%
Single	17.2
Married	52.6
Living with partner	2.2
Separated	2.5
Divorced	15.8
Widowed	5.2
Decline to state	4.5
Ethnic	
Latino	6.5
Non-Latino	85.6
Decline to state	7.9
Race	
American Indian or Alaskan native	0.3
Asian	4.9
Black or African American	11.7
Native Hawaiian or other Pacific Islander	1.6
White	72.8
More than one race	1.1
Decline to state	7.6
Cardiac Dysfunction etiology	
Myocarditis	0.5
Hypertrophic	10.6
Myocardial infarction	7.9
Alcohol abuse	1.6
Idiopathic	2.2
Drug abuse	0.8
Ischemic	3.3
Hypertension	65.4
Valvular dysfunction	10.6
Other	4.4
Left ventricular ejection fraction (mm Hg)	64.4 (9.47) *
Heart rate (bpm)	64.0 (11.9) *

Note: sex, age and BMI are informed in Table 2 with more detail. * [M (SD)].

**Table 2 healthcare-08-00129-t002:** Descriptive analysis of the variables.

Variables		Religious Affiliation		Spirituality		Values
		No	Yes	Low	High	Total
Sex	Men	194	111	49	263	305–312
	Women	6	9	1	14	15
Total sample		200 (62.5%)	120 (37.5%)	50 (15.3%)	277 (84.7%)	320–327 ***
Age [M(SD)]		65.80 (9.50)	66.17 (11.47)	65.32 (9.31)	66.35 (10.48)	66.19 (10.31)
BMI [M(SD)]		30.24 (5.67)	29.71 (4.57)	30.51 (5.06)	29.98 (5.33)	30.06 (5.29)
POS_A [M(SD)]		32.29 (7.61)	33.31 (7.32)	23.80 (6.68) *	34.30 (6.34) *	32.67 (7.43)
NEG_A [M(SD)]		16.18 (6.59)	17.13 (7.91)	24.38 (8.62) *	15.10 (5.77) *	16.54 (7.12)
BDI [M(SD)]		8.23 (7.11)	9.04 (7.76)	18.50 (8.21) *	6.78 (7.32) *	8.56 (7.32)
STAI_S [M(SD)]		30.38 (10.40)	32.12 (11.86)	43.60 (13.70) *	28.56 (8.51) *	31.04 (11.04)
STAI_T [M(SD)]		32.60 (10.66)	35.32 (12.44)	49.60 (11.90) *	30.48 (8.51) *	33.54 (11.50)
AX/EX [M(SD)]		18.77 (9.20)	18.17 (9.70)	27.56 (9.67) *	16.75 (8.40) *	18.47 (9.47)
LOT [M(SD)]		15.39 (4.25)	15.65 (4.02)	10.89 (3.77) *	16.33 (3.61) *	15.52 (4.11)
SWLS [M(SD)] **		23.54 (7.75)	23.75 (7.33)	14.90 (6.94) *	25.31 (6.57) *	23.59 (7.66)
MFSI [M(SD)]		11.65 (19.74)	13.53 (23.42)	39.06 (21.25) *	7.12 (17.14) *	12.14 (21.28)
PSQI [M(SD)]		13.56 (4.10)	12.85 (4.37)	9.96 (3.87) *	13.95 (3.95) *	7.66 (4.19)
CSEQ [M(SD)]		16.27 (4.91)	15.90 (5.03)	10.60 (3.79) *	17.19 (4.45) *	16.18 (4.96)

Note: Only global or total measures and descriptive variables are here resumed: BMI, Body Mass Index; POS_A, Positive Affect Scale; NEG_A, Negative Affect Scale; BDI, Beck Depression Inventory; STAI_S, State Anxiety Inventory; STAI_T, Trait Anxiety Inventory; AX/EX, Anger Expression Index; LOT, Life Orientation Test; SWLS, Satisfaction With Life Scale; MFSI, Multidimensional Fatigue Symptom Inventor; PSQI, Pittsburg Sleep Quality Index; CSES, Cardiac Self-Efficacy Scale. * Oneway ANOVA (*p* < 0.001). ** Sample for this variable is smaller (*n* = 127) due to missing data. *** Seven persons did not respond to the religious affiliation question.

**Table 3 healthcare-08-00129-t003:** The effect of spirituality and religious affiliation on clinical variables.

Predictor	Dependent Variable	Sum of Squares	df	Mean Square	F	*p* <	Partial *η^2^*	Observed Power *
Spirituality	BDITot	4289.88	1	4289.88	117.77	0.001	0.28	1.00
	BDICog	2242.28	1	2242.28	123.11	0.001	0.29	1.00
	BDISom	329.22	1	329.22	46.16	0.001	0.13	1.00
	STAI_S	8210.98	1	8210.98	95.70	0.001	0.27	1.00
	STAI_T	11,764.40	1	11,764.40	139.91	0.001	0.36	1.00
Religious affiliation	STAI_S	860.59	1	860.59	10.03	0.01	0.03	0.88
	STAI_T	844.27	1	844.27	10.04	0.01	0.03	0.88
Spirituality x Religious affiliation	STAI_S	420.07	1	420.07	4.89	0.05	0.01	0.59
Covariate (age):	BDICog	82.21	1	82.21	4.51	0.05	0.01	0.56
	STAI_S	421.89	1	421.89	4.91	0.05	0.02	0.59
	STAI_T	694.67	1	694.67	8.26	0.01	0.03	0.81

Note: Only significative differences are shown. BDITot, depression global score; BDICog, depression cognitive-affective dimension; BDISom, depression somatic dimension; STAI_S, state anxiety; STAI_T, trait anxiety. * Confidence Interval = 0.05.

**Table 4 healthcare-08-00129-t004:** The effect of spirituality and religious affiliation on emotional variables.

Predictor	Dependent Variable	Sum of Squares	df	Mean Square	F	*p* <	Partial *η^2^*	Observed Power *
Spirituality	PosAf	3393.10	1	3393.10	81.79	0.001	0.21	1.00
	NegAf	3726.59	1	3726.59	97.36	0.001	0.24	1.00
	AxIn	935.76	1	935.76	70.21	0.001	0.19	1.00
	AxOut	285.58	1	285.58	32.47	0.001	0.09	1.00
	AxCon	470.40	1	470.40	22.37	0.001	0.07	0.99
	AxFx	4785.66	1	4785.66	63.91	0.001	0.17	1.00
Religious affiliation	NegAf	554.00	1	554.00	14.47	0.001	0.04	0.96
	AxOut	52.89	1	52.89	6.01	0.05	0.02	0.68
Spirituality x Religious affiliation	NegAf	367.10	1	367.10	9.59	0.01	0.03	0.87
	AxOut	114.97	1	114.97	13.07	0.001	0.04	0.95
Covariate (age):	NegAf	427.91	1	427.91	11.18	0.01	0.03	0.91

Note: Only significative differences are shown. PosAf, Positive Affect; NegAf, Negative Affect; AxIn, Anger Expression-In; AxOut, Anger Expression-Out; AxCon, Anger Control; AxFx, Anger Expression Index. * Confidence Interval = 0.05.

**Table 5 healthcare-08-00129-t005:** The effect of spirituality and religious affiliation on physical health variables.

Predictor	Dependent Variable	Sum of Squares	df	Mean Square	F	*p* <	Partial *η^2^*	Observed Power *
Spirituality	GeneralF	1550.80	1	1550.80	49.60	0.001	0.14	1.00
	PhysicalF	847.86	1	847.86	38.72	0.001	0.11	1.00
	EmotF	2676.27	1	2676.27	153.29	0.001	0.34	1.00
	MentalF	1391.60	1	1391.60	80.82	0.001	0.21	1.00
	Vigor	1585.75	1	1585.75	93.69	0.001	0.24	1.00
	TotalF	38,949.68	1	38,949.68	121.79	0.001	0.29	1.00
	SubSQ	6.89	1	6.89	11.54	0.01	0.03	0.92
	SlLat	19.52	1	19.52	18.72	0.001	0.06	0.99
	SlDist	8.91	1	8.91	22.48	0.001	0.07	0.99
	DDysf	25.48	1	25.48	50.60	0.001	0.15	1.00
	SEff	5.50	1	5.50	4.06	0.05	0.01	0.52
	SlDur	5.12	1	5.12	5.13	0.05	0.01	0.61
	PSQITot	509.15	1	509.15	31.94	0.001	0.10	1.00
Religious affiliation	PhysicalF	100.99	1	100.99	4.61	0.05	0.01	0.57
	EmotF	219.94	1	219.94	12.59	0.001	0.04	0.94
	MentalF	88.62	1	88.62	5.14	0.05	0.01	0.61
	TotalF	1420.93	1	1420.93	4.44	0.05	0.01	0.55
	DDysf	4.41	1	4.41	8.75	0.01	0.03	0.83
Spirituality x Religious affiliation	EmotF	94.29	1	94.29	5.40	0.05	0.01	0.63
Covariates: (age)	SubSQ	2.52	1	2.52	4.22	0.05	0.015	0.53
	SlDur	7.58	1	7.58	7.59	0.01	0.026	0.78
	PSQITot	89.50	1	89.50	5.61	0.05	0.019	0.65
(BMI)	PhysicalF	199.90	1	199.90	9.13	0.05	0.030	0.85
	TotalFatigue	1247.68	1	1247.68	3.90	0.05	0.013	0.50
	DDysf	4.06	1	4.06	8.06	0.01	0.027	0.80

Note: Only significative differences are shown. GeneralF, General fatigue; PhysicalF, Physical fatigue; EmotF, Emotional fatigue; MentalF, Mental fatigue; TotalF, Total fatigue; SubSQ, subjective sleep quality; SlLat, sleep latency; SlDist, sleep disturbances; DDysf, daytime dysfunction; SEff, sleep efficiency; SlDur, sleep duration; PSQITot, PSQI global score. * Confidence Interval = 0.05.

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
