# Peer review of "Is Belonging to a Religious Organization Enough? Differences in Religious Affiliation Versus Self-Ratings of Spirituality on Behavioral and Psychological Variables in Individuals with Heart Failure"

_healthcare, 2020, doi:10.3390/healthcare8020129_

Round 1

Reviewer 1 Report

This study addresses an important topic, as spirituality can contribute positive (resilience, self-confidence) as well as negative (i.e. heart diesease as punishment of good) aspects. 

However, there are 3 aspects that have to be addressed:

  1. as the patient sample comprises mostly men, no evaluation of potential gender-specific differences is possible. The authors mention that but it should be discussed a little bit more deeply, which aspects (i.e. gender-differences in prevalence of depression/anxiety) would be important there
  2. at least basic clinical data of the patient sample have to be shown (age, somatic comorbidities, NYHA class, education, social status etc.), in text form or (better) in a table
  3. some of the factors mentioned in 2. should be considered as potential confounding factors in the analyses: NYHA class (depicting the clinical severity and symptom load of heart failure), level of education, social Status, living alone or in partnership. It might well be, that spirituality plays a different role in different heart failure states and it is known that education and social status influence the frequency of depressive and anxious comorbvidities.

Please add information about these aspects and the validity of your data will win a lot!

Thank you

Author Response

Response to Reviewer 1 Comments

Point 1: as the patient sample comprises mostly men, no evaluation of potential gender-specific differences is possible. The authors mention that but it should be discussed a little bit more deeply, which aspects (i.e. gender-differences in prevalence of depression/anxiety) would be important there.

Response 1: Thank you for your constructive advice for improving our study. Following the reviewer recommendations, we have introduced in the Discussion section three new references and provided more details to the gender issue (343-347).

Point 2: at least basic clinical data of the patient sample have to be shown (age, somatic comorbidities, NYHA class, education, social status etc.), in text form or (better) in a table.

Response 2: Thank you for your constructive advice that improve our materials and methods section. Following the reviewer recommendations, we have introduced the new Table 1 (line 80). This Table shows basic sociodemographic and clinical data of the patient sample.

Point 3: some of the factors mentioned in 2. should be considered as potential confounding factors in the analyses: NYHA class (depicting the clinical severity and symptom load of heart failure), level of education, social Status, living alone or in partnership. It might well be, that spirituality plays a different role in different heart failure states and it is known that education and social status influence the frequency of depressive and anxious comorbidities.

Response 3: Thank you for your constructive advice that improve our analysis and results. In Table 2 we had already shown that there was no difference in spirituality considering age, sex and BMI in our sample, but we agree with the referee that there might be other potential confounding factors. So, according to the referee’s advice, we tested the distribution of spirituality and religious affiliation in the basic sociodemographic and clinical data of the patient sample to find any potential confounding factors in the analyses (line 189-191, 212-214). We did not find differences for our sample.

Reviewer 2 Report

Interesting topic. Presented well.

Author Response

Response to Reviewer 2 Comments

We are very grateful for the great impression that our manuscript made to reviewer 2. We went through the text again and, considering also reviewer 3, we made minor typo-like corrections.

Reviewer 3 Report

This is a fine and well written piece of research in my opinion. It clearly contributes to the studies on religiosity/ spirituality in physical and mental health, here esp. in individuals with HF. It deserves nimble publication after minor revision. Congratulations!

I presuppose that the statistics are well done – eventually, a specialist might check for that.

I have one major and two minor observations concerning your discussion of results.

The first one consists in a necessary contextualization or at least attention within former research and concepts:

296-297 you write: “In fact, we observed instances where religious affiliation, when it is not accompanied by a parallel spiritual life, appeared counterproductive.”

In addition to your appropriate use of Festinger’s theory of cognitive dissonance, please put your results into context of former research, which already exists from the early 1960s by Allport(e.g. Allport & Ross 1968) on “intrinsic and extrinsic religiosity”. Your whole paper can also be considered a specific extension of this classic distinction, showing its validity also in the context of “health effects”. Actually, this could even be part of the introduction already.

The minor ones:

291-292 you say that your results contradict others [46], “for which spirituality was negatively related with anger expression” – do your results not show the same inverse correlation: the more spiritual, the less anger expression? – please check and clarify or correct this.

Finally: I think it is important that you finally put into relief what was not as clear before:

326-328: “especially for the people low in spirituality that also belonged to a religious organization, which showed great interactions and was small.”

Do you have any idea for this?? Please add a sentence or two for clarification (again, to me this seems to related to my first observation; cf. Allport).

Some typo-like corrections:

In a few places you write “spiritualty”

  1. 84: insert “they”: they have already …
  2. 94: phenomenon rather than phenomena
  3. 122: 31-49 rather than 31-40?
  4. 143: 32- reference 32 – Stacy, S. – seems incomplete –Stacy’s dissertation e.g. is of 2001.
  5. 155: “to 6” rather than “2”
  6. 170-171 – check the sentence/ grammar
  7. 236-237 the higher THE positive affect score
  8. 266 – do you mean most or more spiritual people?
  9. 283 cancel “of”
  10. 314 at least “in” part
  11. 339: cancel “a”

Author Response

Response to Reviewer 3 Comments

Point 1:  The first one consists in a necessary contextualization or at least attention within former research and concepts:

296-297 you write: “In fact, we observed instances where religious affiliation, when it is not accompanied by a parallel spiritual life, appeared counterproductive.”

In addition to your appropriate use of Festinger’s theory of cognitive dissonance, please put your results into context of former research, which already exists from the early 1960s by Allport(e.g. Allport & Ross 1968) on “intrinsic and extrinsic religiosity”. Your whole paper can also be considered a specific extension of this classic distinction, showing its validity also in the context of “health effects”. Actually, this could even be part of the introduction already.

Response 1: Thank you for your constructive advice that improve our theoretical frame and our Discussion. This comment allows us to introduce Allport’s and his followers´ works. We agree with the referee about Allport´s relevance, so we have dedicated nine lines and introduced 6 new references for this issue (line 313-321).

Point 2:  291-292 you say that your results contradict others [46], “for which spirituality was negatively related with anger expression” – do your results not show the same inverse correlation: the more spiritual, the less anger expression? – please check and clarify or correct this.

Response 2: We apologize for this mistake. We have changed the word “negative” for “positive”… (line 297).

Point 3:  Finally: I think it is important that you finally put into relief what was not as clear before:

326-328: “especially for the people low in spirituality that also belonged to a religious organization, which showed great interactions and was small.”

Do you have any idea for this?? Please add a sentence or two for clarification (again, to me this seems to related to my first observation; cf. Allport).

Response 3: Thank you for your constructive advice. This is really a challenge, but we followed referees´ advice and went further with a couple of ideas… (line 340-342).

Point 4: Some typo-like corrections:

In a few places you write “spiritualty”

  1. 84: insert “they”: they have already … (86)
  2. 94: phenomenon rather than phenomena (96)
  3. 122: 31-49 rather than 31-40? (124)
  4. 143: 32- reference 32 – Stacy, S. – seems incomplete –Stacy’s dissertation e.g. is of 2001. (470)
  5. 155: “to 6” rather than “2” (157)
  6. 170-171 – check the sentence/ gramar (172-173)
  7. 236-237 the higher THE positive affect score (242)
  8. 266 – do you mean most or more spiritual people? (271)
  9. 283 cancel “of” (288)
  10. 314 at least “in” part (327)
  11. 339: cancel “a” (358)

Response 4: Thank you so much for detecting and helping with these errors. We really appreciate referees´ help. We have revised and correct all of them as requested. We write up here the numbers of the new lines to find them easily.